# The Establishment of Thermal Conductivity Model for Linear Low-Density Polyethylene/Alumina Composites Considering the Interface Thermal Resistance

**DOI:** 10.3390/polym14051040

**Published:** 2022-03-05

**Authors:** Guo Li, Yanghui Wang, Huihao Zhu, Yulu Ma, Huajian Ji, Yu Wang, Tao Chen, Linsheng Xie

**Affiliations:** 1School of Mechanical and Power Engineering, East China University of Science and Technology, Shanghai 200237, China; liguo119804@126.com (G.L.); wyh3861594@163.com (Y.W.); zhhxhhy1994@126.com (H.Z.); myl@ecust.edu.cn (Y.M.); hjji@ecust.edu.cn (H.J.); wangyu_ecust@ecust.edu.cn (Y.W.); 2Faculty of Chemical Engineering, Kunming University of Science and Technology, Kunming 650031, China; chentao716@163.com

**Keywords:** thermal conductivity prediction model, thermally conductive composites, Al_2_O_3_, interface thermal resistance

## Abstract

An optimized thermal conductivity model of spherical particle-filled polymer composites considering the influence of interface layer was established based on the classic series and parallel models. ANSYS software was used to simulate the thermal transfer process. Meanwhile, linear low-density polyethylene/alumina (LLDPE/Al_2_O_3_) composites with different volume fractions and Al_2_O_3_ particle sizes were prepared with the continuous mixer, and the effects of Al_2_O_3_ particle size and volume fraction on the thermal conductivity of the composites were discussed. Finally, the test result of the thermal conductivity was analyzed and compared with ANSYS simulations and the model prediction. The results proved that the thermal conductivity model considering the influence of the interface layer could predict the thermal conductivity of LLDPE/Al_2_O_3_ composites more precisely.

## 1. Introduction

Thermally conductive composites have been widely applied in the fields of lighting, solar energy, electronic, medical and health [1,2]. The most common method for preparing polymer composites with desired thermal conductivity is blending the melt polymer matrix with thermally conductive fillers, such as graphite, alumina, aluminum nitride, silicon nitride, boron nitride, etc. [3,4]. The thermal conductivity of polymer composites often depends on the thermal conductivity of filler, as well as the filling content, particle size, surface geometry and microscopic morphology of the filler [5,6,7]. The establishment of the mathematical model on thermal conductivity for polymer composites is of great significance in analyzing the influencing factors of heat transfer performance, revealing the heat transfer mechanism, predicting the thermal conductivity of polymer composites and further optimizing the formulation design and products.

To explain the thermal conduction mechanism of polymer composites, many physical models describing the thermal transfer process for polymer was used, such as thermal conduction path theory [8,9,10], thermal conduction percolation model [11,12], series model and parallel model [9,13,14,15], etc. Moreover, more and more mathematical models for predicting the thermal conductivity of polymer composites have been established based on the shape, size and content of thermally conductive fillers. For example, the Rayleigh classic model, Springer-Tsai model [16] and Agari model [17] apply to fibrous filler-filled polymers. The Hatta model [18] and the Zhai model [19] apply to planar filler-filled polymers. As composites with granular fillers are common in use, there remain some models to predict their thermal conductivity such as the Maxwell model [20], Bruggeman model [21], Russell model [22], Agari model [9], Liang-Liu model [13,14,15], etc. At the same time, some scholars use numerical simulation methods to study the thermal transfer process and simplify the whole composites to the unit body [23,24,25]. These models generally assume that the two-phase interface is infinitely thin, and the effect of the interface phase on the thermal conductivity of the polymer composites could thus be neglected during model establishment or numerical simulation [13,14,15,20,21,22,23,24,25]. However, such an assumption could lead to the visible error between theoretical calculation and experiment. Meanwhile, the fillers are considered to be isolated in the polymer separately in these models. In fact, the fillers will form agglomeration or come into contact with each other in the polymer composites with the increase in filler volume fraction. The thermally conductive network structure is formed when the volume fraction reaches a critical value, which causes deviations in the predicted results.

Some studies have shown that there is an obvious phase region between the filler particles and the polymer matrix, which is called the interface layer [26,27,28], and they offer various methods to measure the thickness of the interface layer. [29,30,31]. In this paper, a new effective thermal conductivity model of spherical particle-filled composites is established based on the classic series and parallel models. The established model considers the existence of an interface layer and its influence on the final thermal conductivity. The ANSYS finite element software is used to simulate the thermal transfer process. At last, a series of LLDPE/Al_2_O_3_ composites were prepared with different Al_2_O_3_ volume fractions and particle diameters to verify the precision of the established thermal conductivity model.

## 2. Experiment and Characterization

### 2.1. Materials and Experiments

Spherical Al_2_O_3_ particles (the diameters of Al_2_O_3_ particle range from 5 μm to 40 μm) are supplied from Zhengzhou Sanhe New Material Co., Ltd (Zhengzhou, China). Linear low-density polyethylene (LLDPE, 1002 KW) with a density of 0.918 g/cm^3^ and melting index of 2 g/10 min is purchased from Suzhou Renfa Plastic Chemical Co., Ltd. (Suzhou, China).

Al_2_O_3_ and LLDPE were vacuum-dried at 80 °C for 8 h. The LLDPE/Al_2_O_3_ composites with different volume fractions were mixed via a two-rotor continuous mixer (rotor diameter is 30 mm). The rotor speed was 600 rpm, and the feed rate was 4000 g/h. The barrel temperature of the solid conveying section and the melt mixing section were 55 °C and 145 °C, respectively. The samples for thermal conductivity measurement were prepared by a plate vulcanizer at 160 °C.

### 2.2. Characterization

Scanning electronic microscopy (JSM-6300LV of JEOL company, Tokyo, Japan) was used to characterize the dispersion and distribution state of Al_2_O_3_ particles in the composites. The scanning electronic microscopy (SEM) samples were fractured in liquid nitrogen and etched in concentrated hydrochloric acid before testing. The etched surfaces were coated in gold before SEM observation. The fractured surface morphologies under different magnifications were recorded.

The laser thermal conductivity testing instrument (LFA447 of NETZSCH company, Bavaria, Germany) was used to obtain the thermal conductivity of the composites. The test samples were molded into a disc with a diameter of 12.7 mm and a height of 1.5~2.0 mm. The upper and lower surfaces of the samples were sprayed with graphite uniformly, and the test temperature is 25 °C.

## 3. Establishment of Thermal Conductivity Model

The SEM images for the fractured surface of LLDPE/Al_2_O_3_ composites are illustrated in Figure 1. Figure 1a,b shows the produced pictures when an average particle size of Al_2_O_3_ was 5 μm, while Figure 1c,d shows the Al_2_O_3_ particles with an average size of about 40 μm. It could be seen that there was clearly a gap between the Al_2_O_3_ particles and the LLDPE matrix. Such gap (also called interface layer or interphase layer) [26] indicated a poor interface between the filler and the matrix, which leads to the appearance of interface thermal resistance and weakens the heat transfer between the matrix and Al_2_O_3_ particles. Fifty different particle images were used to calculate the equivalent average radius ($\bar{r}$) of Al_2_O_3_ particles and the equivalent average radius ($\bar{r_{C}}$), including the thickness of interface layer. $\bar{r}$ and $\bar{r_{C}}$ were calculated by statistical analysis of the SEM images through the Image Pro Plus software [32]. As shown in Figure 1b,d, it can be seen that the interface layer was thicker when Al_2_O_3_ particles increased from 5 μm to 40 μm, and the ratio of $\bar{r_{C}}$ to $\bar{r}$ slightly increased from 1.049 to 1.056.

We assumed that the Al_2_O_3_ particles were spheres and uniformly dispersed in the LLDPE matrix, as illustrated in Figure 2a. The polymer composites comprised numerous thermally conductive units. Each unit was regarded as a cube, which was composed of a separate spherical Al_2_O_3_ particle wrapped in the LLDPE matrix. According to the law of minimum thermal resistance, the law of equivalent thermal conductivity, and the theory of homogenization [13,14], the equivalent thermal conductivity of polymer composites could be regarded as the equivalent thermal conductivity of the unit with the same specific equivalent thermal resistance. In Figure 1, there was an interface layer between the Al_2_O_3_ particle and the LLDPE matrix. Therefore, an interface layer settled around the spherical Al_2_O_3_ particle in the unit. The model for a single unit was established, as shown in Figure 2b. The transfer direction of thermal flux was illustrated in Figure 2b, where the flux came from the upper surface of the unit and then passed through the polymer matrix, the interface layer and the filler particle in sequence. Finally, the flux went out from the lower surface.

According to Fourier’s law:(1)$$Q=\lambda S\frac{\Delta T}{H}=\frac{\Delta T}{R}$$
(2)$$R=\frac{H}{\lambda S}$$

In the equation, *Q* was the thermal flow transferred per unit time; λ was the thermal conductivity of the unit (W/m∙K); *S* was the heat transfer surface vertical to the flow direction of heat flux; ΔT was the temperature difference; *h* was the thickness of the unit; *R* was the total thermal resistance of the unit.

The thermal conductivity model based on the interface thermal resistance was further established, as shown in Figure 3. The unit in Figure 2b was cut into three regions in Figure 3a: upper, middle, and lower regions. The upper and lower regions were the matrix. The middle region consisted of Al_2_O_3_ particles and the matrix, and the thickness of the middle region was 2 r (r is the radius of spherical Al_2_O_3_ particle), as Figure 3c illustrates. Thus, the thermal resistance of the unit could be regarded as the thermal resistance of the three regions in series. Considering that the upper and lower regions were both the LLDPE matrix, the thermal resistance of upper and lower regions were the same and could be treated as region 1. As for the middle region, it was named region 2. Meanwhile, we named the thermal resistance of region 1 and region 2 *R*_1_ and *R*_2_, and the thickness of the unit was *H*.

The thermal conductivity model of region 2 (middle region) was further divided into five regions using a similar method, as illustrated in Figure 3a. The division results were illustrated in Figure 3b, and we assumed the thermal resistances of the three regions were *R*_3_, *R*_4_, and *R*_5_*,* respectively. Figure 3c shows a cross-sectional view of region 4.

According to Equation (2), the thermal resistance of each region obtained above could be calculated with the following equations.

Region 1:(3)$$R_{1}=\frac{H-2r}{2\lambda_{p}S}=\frac{H-2r}{2\lambda_{p}H^{2}}$$

Region 2:(4)$$\frac{1}{R_{2}}=\frac{1}{R_{3}}+\frac{1}{R_{4}}+\frac{1}{R_{5}}$$

Region 3:(5)$$R_{3}=\frac{2r}{\lambda_{p}(H^{2}-2rH)}$$

Region 4:

Considering that region 4 contained the thermal resistance of the Al_2_O_3_ particle, LLDPE, and interface layer, which were marked with $R_{f}$, $R_{P}$, and $R_{C}$, respectively. The constant *C* was introduced as the ratio of $R_{C}$ to $R_{f}$.
(6)$$C=\frac{R_{c}}{R_{f}}$$

According to the similarity of thermal conduction and electrical conduction, the thermal resistance of Al_2_O_3_ particles and the interface layer was equivalent to the $R_{C}$ series with $R_{f}$ marked with $R_{IC}$:(7)$$R_{IC}=R_{f}+R_{C}=(C+1)R_{f}$$

For the spherical particle, the particle was cut into small elements with thickness dy along the y direction, as shown in Figure 3c. Thus, $R_{4}$ was calculated by integrating and can be expressed as:(8)$$R_{4}=\int_{0}^{2r}(\lambda_{p}\frac{S_{p}}{S}+\lambda_{f}\frac{S_{f}}{S})dy=\frac{2r}{\lambda_{p}(4r^{2}-\frac{2\pi r^{2}}{3})+\lambda_{f}\frac{2\pi r^{2}(C+1)}{3}}$$

Region 5:(9)$$R_{5}=\frac{2r}{\lambda_{p}(2rH-r^{2})}$$

Based on the equations above, the total thermal resistance of region 2 could be obtained through Equation (4):(10)$$R_{2}={\left[\frac{\lambda_{p}(H^{2}-2rH)+\lambda_{p}(4r^{2}-\frac{2\pi r^{2}}{3})+\lambda_{f}\frac{2\pi r^{2}(C+1)}{3}+\lambda_{p}(2rH-r^{2})}{2r}\right]}^{-1}$$

In the equation, *H* is the thickness of the unit; *r* is the radius of the Al_2_O_3_ particle; $\lambda_{p}$ and $\lambda_{f}$ represent the thermal conductivity of the LLDPE matrix and Al_2_O_3_ particles, respectively, (W/m∙K); $R_{1}$, $R_{2}$, $R_{3}$, $R_{4}$, and $R_{5}$ are the thermal resistance for region 1 to region 5, respectively, (K/W); the constant *C* is the ratio of $R_{C}$ to $R_{f}$; $R_{f}$ and $R_{C}$ are the thermal resistance of the Al_2_O_3_ particles and the interface layer, respectively, (K/W).

According to the equivalent model, the volume fraction of filler in composites was:(11)$$\phi_{f}=\frac{4\pi r^{3}}{3H^{3}}$$

The constant α was a coefficient related to the volume fraction of filler:(12)$$\alpha =\frac{r}{H}=(\frac{3\phi_{f}}{4\pi }{)}^{\frac{1}{3}}$$

The effective thermal conductivity of the unit could be simplified as:(13)$$\lambda_{eff}={\left\{\frac{1}{\lambda_{p}}(1-2\alpha )+{\left[(\frac{1}{2\alpha }-\frac{\alpha }{2})\lambda_{p}+(2\alpha -\frac{\pi \alpha }{3})\lambda_{p}+\frac{\pi \alpha }{3(C+1)}\lambda_{f}\right]}^{-1}\right\}}^{-1}$$

When ignoring the thermal resistance of the interface layer (*C* = 0), Equation (14) becomes:(14)$$\lambda_{eff}={\left\{\frac{1}{\lambda_{p}}(1-2\alpha )+{\left[(\frac{1}{2\alpha }+\frac{3\alpha }{2}-\frac{\pi \alpha }{3})\lambda_{p}+\frac{\pi \alpha }{3}\lambda_{f}\right]}^{-1}\right\}}^{-1}$$

## 4. Finite Element Simulation for Thermal Transfer Process

### 4.1. Model and Boundary Conditions

The simulation model for a thermally conductive unit is presented in Figure 4 according to the physical model shown in Figure 2. The thermal resistance of the interface layer between the spherical Al_2_O_3_ particle and the LLDPE matrix was neglected in Figure 4a. The thermal resistance and thickness of the interface layer were considered in Figure 4b. The finite element software ANSYS was used to simulate the thermal transfer process of the unit. In both cases, the steady-state model was chosen. The thermal flux input from the upper surface of the unit, as shown in Figure 4, and the four walls of the unit were set to the adiabatic boundary condition. The lower surface was natural air convection, and the air convection temperature was 25 °C. Constant heat flowed into the unit from the upper surface and flowed out of the lower surface. The average particle size of Al_2_O_3_ particles was obtained from SEM images, as presented in Figure 1, and the wall height of the unit (*H*) was calculated according to Equation (12).

### 4.2. Simulation Parameter

Based on the thermal differential equation of the spherical shell, the thermal resistance of the interface layer was integrated into Equation (15):(15)$$R_{C}=\frac{\frac{1}{\bar{r}}-\frac{1}{\bar{r_{c}}}}{4\pi \lambda_{c}}=\frac{\bar{r_{c}}-\bar{r}}{4\pi \lambda_{c}\bar{r_{c}}\bar{r}}$$

In the equation, $\lambda_{C}$ is the thermal conductivity of interface layer (W/m∙K).

We assumed that the interface layer between the Al_2_O_3_ particle and the LLDPE matrix was filled with air. Under such assumption, the thermal resistance of the interface layer was regarded as the thermal conductivity of air at room temperature (25 °C). The parameters were as follows:

LLDPE: density was 0.918 g/cm^3^; thermal conductivity at 25 °C was 0.27 W/(m∙K);

Al_2_O_3_ particles: density was 3.7 g/cm^3^; thermal conductivity at 25 °C was 30 W/(m∙K);

Interface layer (air): density was 1.146 × 10^−3^ g/cm^3^; thermal conductivity at 25 °C was 2.552 × 10^−2^ W/(m∙K).

### 4.3. Simulation Results and Discussions

Figure 5 shows the temperature contour and thermal flux vector graph of the thermally conductive unit for LLDPE/Al_2_O_3_ composites when the volume fraction of Al_2_O_3_ was 9.8% and the average particle size of Al_2_O_3_ was 5 μm. Figure 5a,b shows the temperature contour and thermal flux vector graph neglecting the interface layer, respectively. Figure 5c,d shows the temperature contour and thermal flux vector graph considering the interface layer, respectively. In Figure 5a, the temperature gradient cannot be observed obviously in the LLDPE matrix, while the temperature gradient in the Al_2_O_3_ particle was not presented. The reason was that the Al_2_O_3_ particle had higher thermal conductivity compared to LLDPE. Thus, the loss of thermal flux was much less when flux flowed through the Al_2_O_3_ particle than through the LLDPE matrix. When considering the interface layer, the unit exhibited a larger temperature gradient in Figure 5c than in Figure 5a. In Figure 5b,d, the arrow expresses the flow direction of thermal flux, and the length and color of the arrow indicate the intensity of the thermal flux. When thermal flux passes through the Al_2_O_3_ particle, the transfer process was greatly enhanced and facilitated due to the higher thermal conductivity of the Al_2_O_3_ particle. Along the flow direction, the transfer process of thermal flux at the contact interface was strong. While perpendicular to the flow direction, the thermal flux at the contact interface was rather weak. This result indicates that the thermal flux was directional. Moreover, the change in thermal flux density was more significant when considering the interface layer, as Figure 5b,d illustrates.

The temperature change after thermal flux passing through the unit was obtained, and the thermal conductivity of the unit with different volume fractions was calculated according to Fourier’s law, which was equivalent to the thermal conductivity of the composites. Eight units were stacked (similar to Figure 2a) to simulate the thermal transfer for LLDPE/Al_2_O_3_ composites. The ANSYS simulation results for effective thermal conductivity are listed in Table 1.

## 5. Experimental Results and Discussions

The morphology of the fractured surface of LLDPE/Al_2_O_3_ composites with different volume fractions is presented in Figure 6. The equivalent average radius ($\bar{r}$) of Al_2_O_3_ particles was 3.25 μm. The Al_2_O_3_ particles were uniformly dispersed in the composites, and Al_2_O_3_ particles were still spheres, as shown in Figure 6. The number of Al_2_O_3_ particles in the LLDPE matrix increased with the volume fractions under the same magnification. When the volume fraction of Al_2_O_3_ particles was 9.8%, the distances among particles were large, and particles were distributed almost isolated in the composites. When the volume fraction of Al_2_O_3_ increased to 27.6%, the distances among particles were significantly reduced, and some of the particles came into contact with each other in the composites. It can be clearly observed that most of the Al_2_O_3_ particles came into contact with each other when the volume fraction of Al_2_O_3_ increased to 32.5%, as shown in Figure 6e. There were many cracks and pores around the Al_2_O_3_ particles (Figure 6b,d,f). The adhesion between the Al_2_O_3_ particle and the LLDPE matrix was weak, and some particles were pulled out when preparing the samples for SEM observation. The reason was that there was an interface layer between the Al_2_O_3_ particle and the LLDPE matrix, and the Al_2_O_3_ particle was not tightly wrapped by the matrix.

The thermal conductivity measurement results for LLDPE/Al_2_O_3_ composites with different volume fractions are illustrated in Figure 7a. It could be seen from Figure 7a that the thermal conductivity of the composites increased when the Al_2_O_3_ volume fraction increased. Meanwhile, the thermal conductivity of the composites showed sharp improvement when the volume fraction of Al_2_O_3_ increased from 27.6% to 32.5%. The phenomenon could be analyzed from Figure 6: The distance among Al_2_O_3_ particles became closer and the Al_2_O_3_ particles were still individually wrapped by LLDPE matrix when the volume fraction of Al_2_O_3_ increased from 9.8% to 27.6%. In this case, the thermal conductivity of the composite slowly rose to an intermediate level and showed an approximate nonlinear relationship with the filler volume fraction. When the Al_2_O_3_ volume fraction increased to 32.5%, most of the Al_2_O_3_ particles came into contact with each other, indicating that a preliminary complete three-dimensional thermal network was formed in the composites. Thus, the thermal conductivity of the composites exhibited a significant increase. Furthermore, the thermal conductivity of the composites continued to increase linearly when the Al_2_O_3_ volume fraction was larger than 32.5%.

Non-metallic fillers mainly rely on adjacent atomic vibrations and lattice heat transfer in ordered crystals [33,34]. When the volume fraction was less than 27.6%, the Al_2_O_3_ particles were individually wrapped by the LLDPE matrix, and the LLDPE/Al_2_O_3_ composites with smaller particle sizes had a higher thermal conductivity because the atomic vibrations of the small Al_2_O_3_ particles were stronger during the thermal transfer process. At the same time, the Al_2_O_3_ particles with a smaller size had a smaller volume and larger specific surface area at the same volume fraction. Additionally, the interface adhesion between Al_2_O_3_ particles and the LLDPE matrix was improved compared to that of the Al_2_O_3_ particles with larger sizes. The improved interface adhesion would alleviate the negative effect of the interface layer on the thermal conductivity of the composites. On the contrary, the interface adhesion between Al_2_O_3_ particles and the LLDPE matrix was weak for Al_2_O_3_ particles with a larger size, and more defects (such as pores and cracks) were generated during the melt blending process. The weak interface adhesion could improve the interface thermal resistance and reduce the thermal conductivity of the composites. The number of Al_2_O_3_ particles increases with the increase in the Al_2_O_3_ volume fraction, which affects the amplitude of atomic vibration. However, the thermal conductivity of the composites had been improved owing to the forming of a thermal conduction path. Additionally, the LLDPE/Al_2_O_3_ composites with larger particle sizes have a higher thermal conductivity because the larger particles tend to come into contact with each other, which makes it easier to form a thicker thermal conduction path. This was the reason that the composites of Al_2_O_3_ particles with a larger size showed a relatively higher thermal conductivity when the volume fraction exceeded 27.6%.

Substituting the measured thermal conductivity into Equation (13), the interface thermal resistance constant *C* was obtained. *C* versus the volume fraction of Al_2_O_3_ particles was presented in Figure 7b. The curves for the thermal resistance constant *C* exhibited a trend of decreasing when the volume fraction of Al_2_O_3_ particles increased. The *C* decreased significantly when the volume fraction increased from 9.8% to 14.5%, and the decreasing trend became slow when the volume fraction increased from 17.2% to 32.5%. The trend finally remained unchanged when the volume fraction grew larger than 32.5%. The reason for this was similar to the change in the thermal conductivity. When the volume fraction of Al_2_O_3_ was lower than 14.5%, the Al_2_O_3_ particles were completely isolated by the LLDPE matrix. Thus, the interface thermal resistance became strong, and the value of C became large. When the volume fraction of Al_2_O_3_ increased from 14.5% to 32.5%, some of the Al_2_O_3_ particles came into contact with each other, forming the thermal conduction path in the composites. Therefore, the interface thermal resistance and the value of C reduced slowly. When the volume fraction of Al_2_O_3_ particles was larger than 32.5%, the dual continuous phase of particles and the LLDPE matrix was formed in the composites [35]. A complete thermal conduction path was formed under this situation, and the interface thermal resistance constant C did not change with the volume fraction anymore.

Comparing the thermal resistance constant *C* of different particle sizes under the same volume fraction in Figure 7b, the interface thermal resistance constant *C* of the Al_2_O_3_ particle with 5 μm was larger when the volume fraction of the Al_2_O_3_ particle was less than 12%. The reason was that the number of Al_2_O_3_ particles with 5 μm was larger under the same volume fraction, and a larger number of Al_2_O_3_ particles were wrapped with the LLDPE matrix, forming a larger interfacial thermal resistance. When the volume fraction of Al_2_O_3_ particles was between 14.5% and 23.7%, the number of particles increased as the volume fraction increased, and Al_2_O_3_ particles with 5 μm were more likely to have local mutual contact and to reduce the interface thermal resistance. Therefore, the thermal resistance constant *C* was slightly smaller than that of the Al_2_O_3_ particle with 40 μm. When the volume fraction of Al_2_O_3_ was more than 27.6%, the particle size had little effect on the interface thermal resistance constant *C* because the thermal conduction path had been initially formed.

Figure 8 illustrates the thermal conductivity of LLDPE/Al_2_O_3_ composites at 25 °C obtained by simulation and experiment. For the simulation results, the thermal conductivity increased linearly with the volume fraction of Al_2_O_3_ particles. Meanwhile, the composites had higher thermal conductivity when the interface layer was ignored. In Figure 8, the curve for simulation results of thermal conductivity was consistent with the curve for experiment results when the volume fraction of Al_2_O_3_ particles increased from 9.8% to 27.6%, while the curve of thermal conductivity obtained via simulation was lower than the curve of the experiment when the volume fraction of Al_2_O_3_ particle exceeded 27.6%. This mainly owed to the decreased distance and the contact between particles when the volume fraction of Al_2_O_3_ particles was over 27.6%, and the thermal conductivity of the composites had been significantly improved because of the forming of thermal conduction path, while it was assumed that the particles were uniformly and separately distributed in LLDPE during the simulation. Such assumption did not consider the influence of the thermal conduction path originating from the particle contact on the thermal conductivity. This was the primary reason causing the inconsistent between simulation and experiment when the volume fraction of Al_2_O_3_ particles was higher than 27.6%.

Figure 9 presented the thermal conductivity calculated from Equations (13) and (14) for the composites with different filling content, and the measured thermal conductivity was also exhibited for comparison. The interface thermal resistance constant in Equation (13) is *C* = 25. In Figure 9, the effective thermal conductivity of the composites according to Equation (13) matched the measured thermal conductivity much better than that using Equation (14), especially when the volume fraction of Al_2_O_3_ exceeded 27.6%. Moreover, Equation (13) was also suitable for predicting the thermal conductivity of LLDPE/Al_2_O_3_ composites with different particle sizes, as Figure 9a,b illustrates.

The thermal conductivity of LLDPE/Al_2_O_3_ composites with different Al_2_O_3_ diameters obtained from the different prediction models, as well as the measured thermal conductivity, was presented in Figure 10 for comparison. Figure 10b,d shows the partially enlarged view for Figure 10a,c, respectively. Figure 10 indicated that the prediction model Equation (13) (considering the interface layer) of thermal conductivity was much more accurate than other models when predicting the thermal conductivity of Al_2_O_3_/LLDPE composites. Furthermore, the thermal conductivity could be predicted more precisely through Equation (13) when the volume fraction of Al_2_O_3_ exceeded 27.6%.

## 6. Conclusions

In this paper, an optimized thermal conductivity model for LLDPE/Al_2_O_3_ composites was established. This model was derived on the basis of the series and the parallel model of thermal conduction considering the interface layer between thermal conduction filler and polymer matrix. Then, the effects of Al_2_O_3_ particle size and volume fraction on the thermal conductivity of LLDPE/Al_2_O_3_composites were studied by experiment and Ansys simulation. The results show that the ANSYS simulation results were more consistent with the experimental value when the volume fraction of Al_2_O_3_ particles in the LLDPE/Al_2_O_3_ composites material was less than 27.6%. When the volume fraction of Al_2_O_3_ particles exceeded 27.6%, the prediction through the model (Equation (13)) was more consistent with the experimental results. This new thermal conductivity model considering the interface layer contributed to a more accurate prediction of the thermal conductivity of the spherical particle-filled polymer.

## Figures and Tables

**Figure 1 polymers-14-01040-f001:**
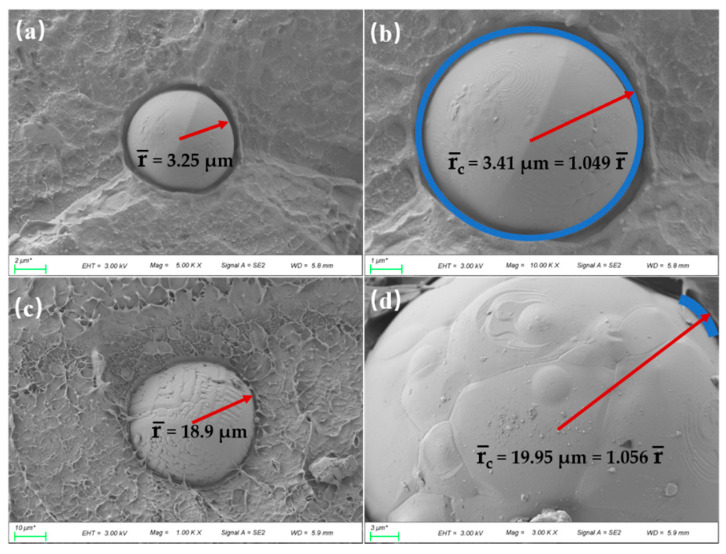
SEM images of LLDPE/Al_2_O_3_ composites with different Al_2_O_3_ diameter: (**a**,**b**) for Al_2_O_3_ diameter with 5 μm; (**c**,**d**) for Al_2_O_3_ diameter with 40 μm.

**Figure 2 polymers-14-01040-f002:**
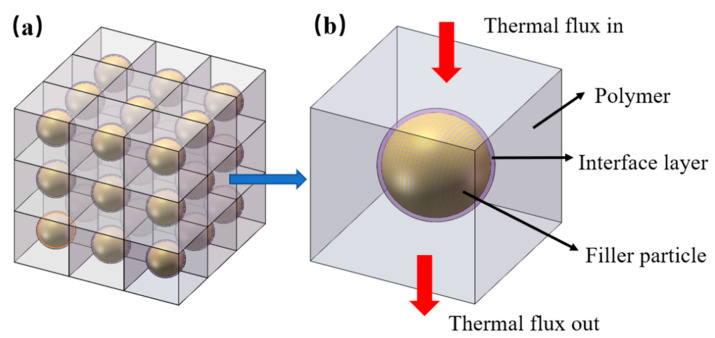
Heat transfer model for LLDPE/Al_2_O_3_ composites: (**a**) heat transfer model composed of numerous thermally conductive units for composites; (**b**) heat transfer model for a single thermally conductive unit.

**Figure 3 polymers-14-01040-f003:**
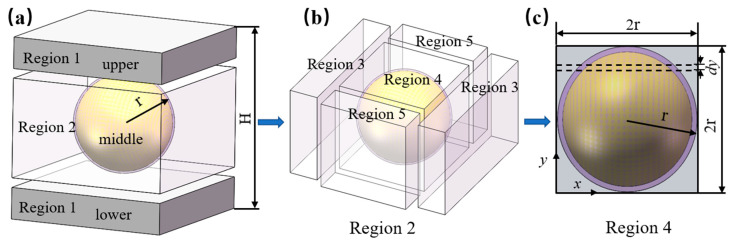
The division strategy for the unit: (**a**) the whole unit; (**b**) region 2; (**c**) a cross-sectional view of region 4.

**Figure 4 polymers-14-01040-f004:**
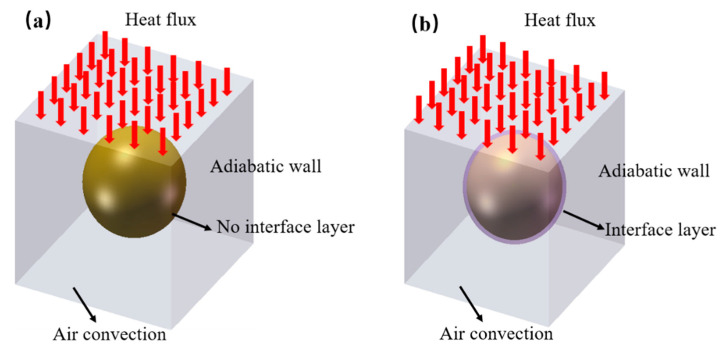
The model of finite element simulation for a thermally conductive unit: (**a**) model without interface layer; (**b**) model with interface layer.

**Figure 5 polymers-14-01040-f005:**
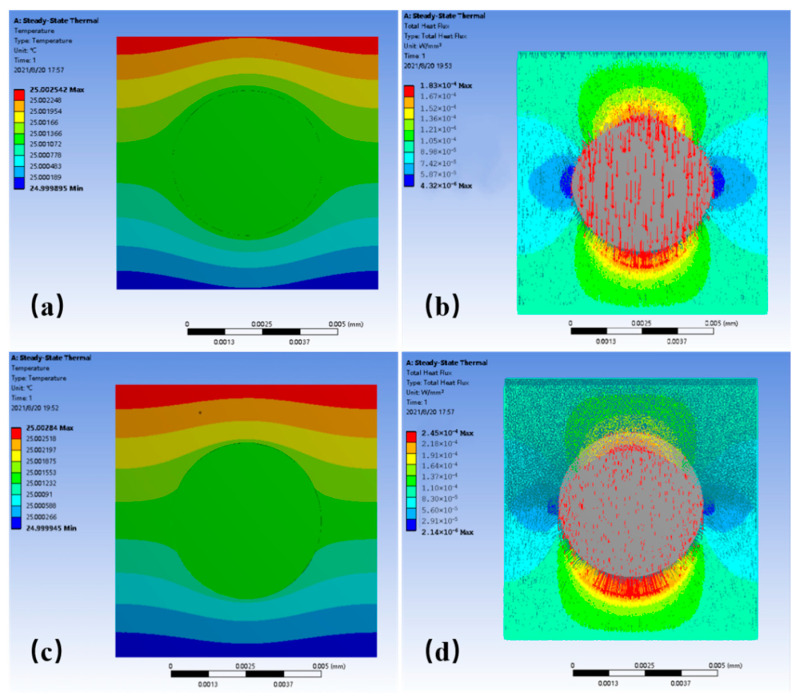
The simulations results of a thermally conductive unit for the LLDPE/Al_2_O_3_ composite when the volume fraction of Al_2_O_3_ was 9.8% and the average particle size of Al_2_O_3_ was 5 μm: (**a**,**c**) the temperature contour; (**b**,**d**) the thermal flux vector graph; (**a**,**b**) without interface layer; (**c**,**d**) with interface layer.

**Figure 6 polymers-14-01040-f006:**
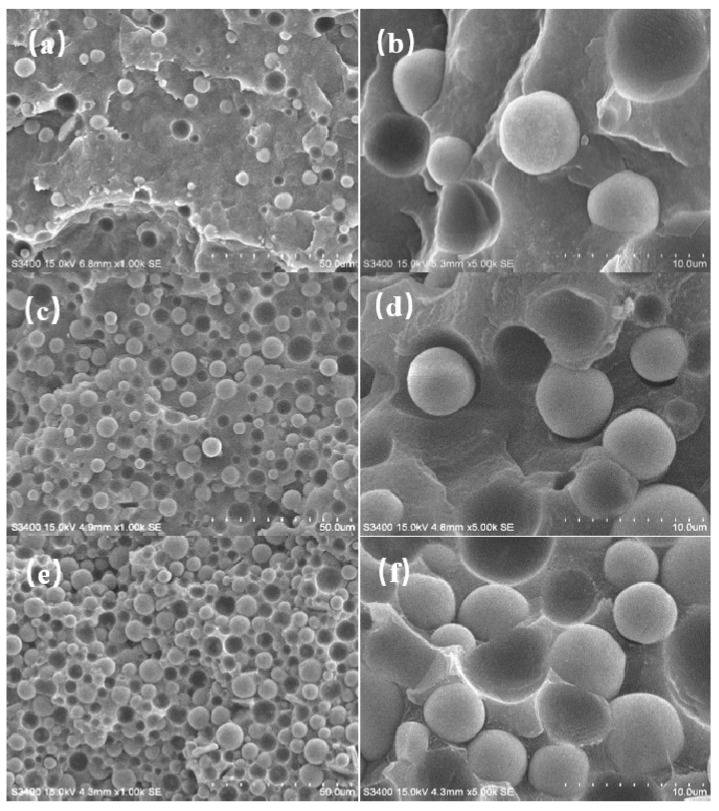
SEM micrographs of LLDPE/Al_2_O_3_ composites with different Al_2_O_3_ volume fractions: (**a**,**b**) 9.8 vol.%; (**c**,**d**) 27.6 vol.%; (**e**,**f**) 32.5 vol.%.

**Figure 7 polymers-14-01040-f007:**
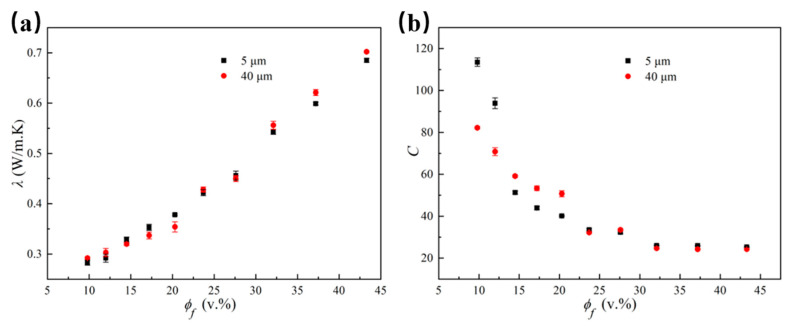
Thermal conductivity and the constant *C* of LLDPE/Al_2_O_3_ composites with two Al_2_O_3_ diameters under different Al_2_O_3_ volume fractions: (**a**) thermal conductivity; (**b**) constant *C.*

**Figure 8 polymers-14-01040-f008:**
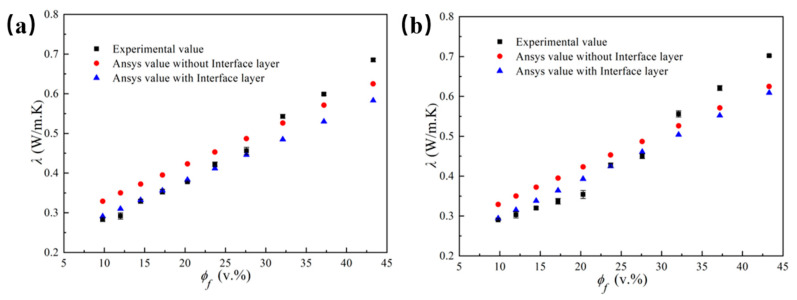
The simulation and experiment results of thermal conductivity for LLDPE/Al_2_O_3_ composites with different Al_2_O_3_ diameters versus volume fractions of Al_2_O_3_ particles: (**a**) for 5 μm; (**b**) for 40 μm.

**Figure 9 polymers-14-01040-f009:**
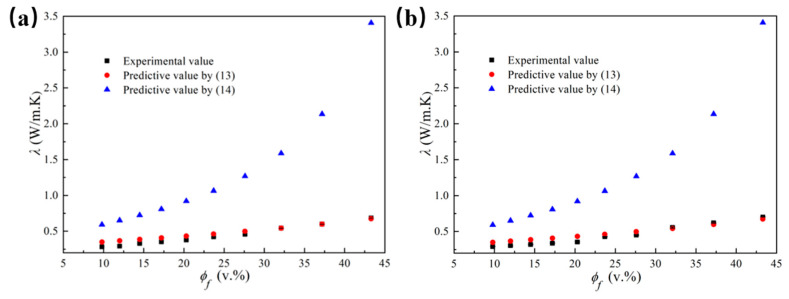
The thermal conductivity of LLDPE/Al_2_O_3_ composites with different Al_2_O_3_ diameters obtained from prediction model and experiment versus the volume fractions of Al_2_O_3_ particles: (**a**) for 5 μm; (**b**) for 40 μm.

**Figure 10 polymers-14-01040-f010:**
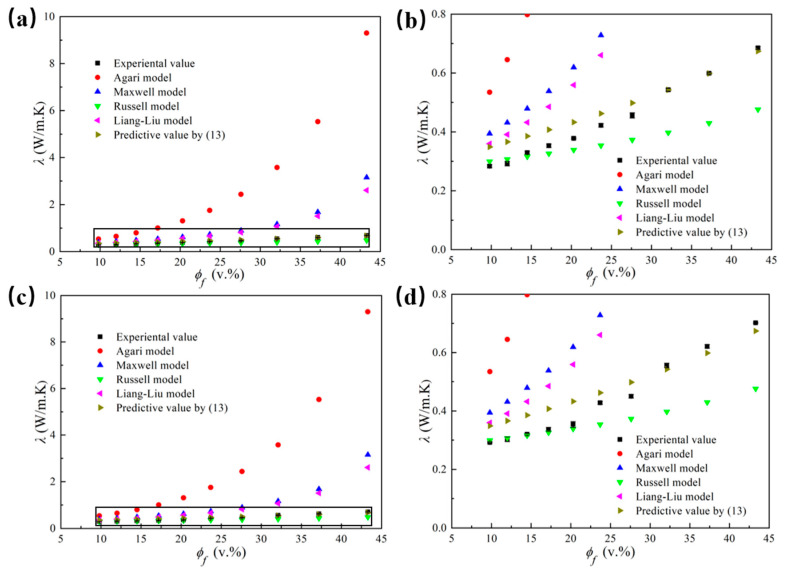
The thermal conductivity of LLDPE/Al_2_O_3_ composites with different Al_2_O_3_ diameters obtained from prediction model and experiment versus the volume fractions of Al_2_O_3_ particles: (**a**,**b**) for 5 μm; (**c**,**d**) for 40 μm.

**Table 1 polymers-14-01040-t001:** The thermal conductivity of LLDPE/Al_2_O_3_ composites with different volume fractions via ANSYS simulations.

Al_2_O_3_ Volume Fractions	$\lambda_{eff}$ without Interface Layer (W/m∙K)	$\lambda_{eff}$ with Interface Layer (W/m∙K)
9.8%	0.329	0.291
12%	0.35	0.31
14.5%	0.372	0.332
17.2%	0.395	0.356
20.3%	0.423	0.383
23.7%	0.453	0.412
27.6%	0.487	0.446
32.1%	0.526	0.485
37.2%	0.571	0.53
43.3%	0.625	0.583

## Data Availability

The data presented in this study are available on request from the corresponding author

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
