# Peer review of "The Establishment of Thermal Conductivity Model for Linear Low-Density Polyethylene/Alumina Composites Considering the Interface Thermal Resistance"

_polymers, 2022, doi:10.3390/polym14051040_

Round 1
Reviewer 1 Report
In this work, the authors presented an optimized thermal conductivity model for LLDPE/Al2O3 composites. The authors studied the effects of Al2O3 particle size and volume fraction on the thermal conductivity of LLDPE/Al2O3 composites using both experiment and simulation
I have provided a minor revision, after answering all the questions listed below, this work can be justified for publishing in Polymers
- Are there any reports on LLDPE/Al2O3 nanocomposites? If so, please write a few sentences to compare the thermal conductivity model for LLDPE/Al2O3 composites and nanocomposites in which the Al2O3 particle size reduced from micron to nano. Will the nanocomposites show better performance?
- Please provide your experimental plots with an error bar
- Instead of connecting data points with lines, repeat the experimental plots in the following manner.
Example: In Fig. 7 (a) and (b), thermal conductivity and C against different Al2O3 volume fractions, either only keep the symbols with error bar (no line), or if you want to connect the data points draw line manually. This method would better present your data. Please do this modification in all the experimental plots.
- It would be great if the authors discuss a few lines about the change in thermal conductivity with different sizes from an atomic point of view (physics of the problem), the authors may explain based on available literature.
Author Response
Dear reviewer:
Thank you for your comments, Those comments are all valuable and very helpful in improving the paper quality.
Please see the attachment is the responses to your comments!

Reviewer 2 Report
This paper presents a model of thermal conductivity of spherical particle-filled polymer matrix composites. The Authors took into account the interfacial layer between the matrix and the fillwe and used the classic series and parallel models as a basis. In general, the paper has been well prepared and deserves publication certain minor revision. Below are my recommendations.
- Please give LLDPE in full both in the title and in the Abstract. This will attract more readers to your work. Not everyone is familiar with this abbreviation.
- Please specify the conditions of gold coating of the specimens before SEM (approximate deposited layer thickness).
- Is it possible to take into account the surface roughness of the Al2O3 particles in the model? If a fraction of the "interface" consists of alumina in the form of asperities, for example?
- Please clarify the following statement: "The small Al2O3 particles would be dispersed in the LLDPE matrix separately at lower volume fraction, and the interface adhesion between Al2O3 particles and LLDPE matrix was improved than that of the Al2O3 particle with larger size". The phrase "separately at lower volume fraction" is not clear. Is the comparison presented here made for composites with the same volume fraction of Al2O3 but different Al2O3 particle sizes?
- Lines 310-314: this paragraph needs revision. 40-micrometer particles are compared with 5-micrometer particles.. This should be made clear. Sentences are not well connected in this paragraph.
- The Conclusions need to be revised: "The results shown that the simulation results were more consistent with the experimental value when the volume fraction of Al2O3 particles in the LLDPE/Al2O3 composites material was less than 27.6%. When the volume fraction of Al2O3 particles exceeded 27.6%, the prediction through the model was consistent with the experimental results". Consistent or inconsistent? It seems that there is a contradiction.
Author Response

(The authors gave the same response as above.)
